# A New Tool to Decrease Interobserver Variability in Biomarker Annotation in Solid Tumor Tissue for Spatial Transcriptomic Analysis

**DOI:** 10.3390/cimb47070531

**Published:** 2025-07-09

**Authors:** Sravya Palavalasa, Emily Baker, Jack Freeman, Aditri Gokul, Weihua Zhou, Dafydd Thomas, Wajd N. Al-Holou, Meredith A. Morgan, Theodore S. Lawrence, Daniel R. Wahl

**Affiliations:** 1Department of Radiation Oncology, University of Michigan, Ann Arbor, MI 48109, USA; palava@umich.edu (S.P.); emibaker@umich.edu (E.B.);; 2Rogel Cancer Center, University of Michigan, Ann Arbor, MI 48109, USA; 3Department of Pathology, Michigan Medicine, University of Michigan, Ann Arbor, MI 48109, USA; 4Department of Neurosurgery, University of Michigan, Ann Arbor, MI 48109, USA

**Keywords:** spatial transcriptomics, immunofluorescent image analysis, interobserver variability

## Abstract

Integrating spatial transcriptomic data with immunofluorescence image data is challenging using existing tools due to their differences in spatial resolution. Immunofluorescence provides information about protein expression at the cellular or subcellular level, whereas spatial transcriptomic platforms typically rely on multicellular “spots” for RNA profiling. Our study coupled spatial transcriptomics of irradiated glioblastoma tissues with immunofluorescence for γH2AX, a marker of DNA damage within the nuclei of cells. We then compared gene expression in γH2AX-positive and negative regions within the tissue. There was significant interobserver variability in manual annotation of γH2AX positivity in multicellular spots by three different researchers (Kappa statistic = 0.345), despite all of them being familiar with γH2AX immunofluorescence and having predefined imaging parameters for annotation. This variability led to different researchers nominating different genes as being associated with DNA repair. To overcome this problem, we have developed a new tool using MATLAB. This tool performs “spot”-wise image analysis and uses researcher-defined parameters such as immunofluorescent marker intensity threshold and number of positive cells to annotate the “spots” as γH2AX positive or negative. The tissue with the most variability in manual annotation was annotated reproducibly by our MATLAB tool, leading to reproducible downstream analysis.

## 1. Introduction

Transcriptomic methods, such as RNA sequencing, allow the study of gene expression in cells and tissue samples [1,2]. Initial transcriptomic technologies interrogated bulk tissues and thus did not address questions pertaining to tissue heterogeneity. Single-cell RNA sequencing enables the analysis of individual cells within a tissue, but it does not address questions about the spatial context, such as cell–cell interactions among neighboring cells. More recently, spatial transcriptomics has emerged, allowing for the spatial analysis of gene expression in intact tissues and thereby enabling a new understanding of how gene expression varies within cell neighborhoods [3]. While spatial transcriptomics is a powerful technology, it does not provide information on post-translational modifications such as phosphorylation or methylation, which can regulate biology independently of gene expression. Therefore, combining spatial transcriptomics with techniques that provide spatial information on post-translational modifications (such as immunofluorescence) can help understand the spatial control of important biological phenotypes [4,5].

Integrating immunostaining data with spatial transcriptomics has numerous challenges, including harmonizing varied degrees of spatial resolution. Some platforms for spatial transcriptomics (such as the Visium and Cytassist platforms from 10x Genomics) permit RNA analysis in “spots”, allowing for analysis of cell groups rather than single cells. These slides typically capture RNA information in spots 55 μM wide and can have up to 5000 spots per capture area [6]. These spots contain variable numbers of cells depending on the tissue type. Immunostaining, by contrast, offers much greater spatial resolution and typically enables the interrogation of single cells. Because immunostaining analysis packages (Cell Profiler 4.2.8, ImageJ 1.54, and QuPath 0.6.0) are optimized for single-cell analysis [7,8], extensive manual processing and annotation (for example, with the LoupeBrowser 10x Genomics tool) are currently needed to integrate immunostaining data with spot-based spatial transcriptomics. These limitations make it difficult to objectively combine immunofluorescence data with spatial transcriptomic data. For example, if the researcher wishes to compare RNA expression in regions with high imaging marker expression with those in regions with low imaging marker expression, the researcher must manually annotate thousands of spots based on the immunofluorescent marker. This manual immunofluorescent spot annotation is both time-consuming and prone to interobserver variability, as the same spot can be annotated differently by different observers. This, in turn, will affect the analysis of spatial RNA expression, and the results of the analysis of the same tissue by different researchers can yield different results, thereby reducing reliability.

To address these limitations, we have designed a new program to integrate spatial transcriptomics and immunofluorescence data using MATLAB R2024b. While this program is generalizable to any immunofluorescent marker, we focused on γH2AX, a histone phosphorylation mark that indicates the presence of DNA damage. Our results indicate that the manual integration of immunofluorescent data and spot-based spatial transcriptomics exhibits high inter-observer variability, which can affect the nomination of biological processes for further study. By developing a new program that automates cell segmentation and annotates “spots” based on quantitative immunofluorescent information, our new analysis tool both reduces inter-observer variability and decreases the time needed for data analysis.

## 2. Materials and Methods

### 2.1. Tissue Sections

The code was generated using brain cancer tissue with radiation-induced DNA damage. Radiation exposure induces double-stranded DNA breaks, which can be detected by immunofluorescence using the phosphorylated Histone marker, γH2AX. This is a post-translational modification that cannot be detected by RNA sequencing. We used mouse brain cancer (glioblastoma) tumor tissue for this study.

### 2.2. Animal Models

All mouse experiments were performed in accordance with the guidelines approved by the Institutional Animal Care and Use Committee (IACUC) at the University of Michigan (approved protocol: PRO00010680). C57BL/6J mice were purchased from Jackson Laboratories. A genetically engineered mouse model, known as the TRP mouse, derived glioblastoma cells were provided as a Kind gift by C. Ryan Miller [9] and were used to generate orthotopic syngeneic glioblastoma tumors, as described previously [10]. In brief, TRP tumor cells (∼5 × 105) were orthotopically implanted in 3 female C57BL/6J mice. Two of the three brain tumor-bearing mice were subjected to 4Gy radiation treatment (RT) to the brain after sedation with 2.5% isoflurane in an Orthovoltage irradiator. One mouse was not irradiated and served as control tumor tissue in this study. The 2 irradiated brain tumor-bearing mice were euthanized through isoflurane overdose followed by cervical dislocation, one at 30 min post-irradiation and the other at 6 h post-irradiation. The unirradiated mouse was euthanized when neurological deficit developed. There was no randomization or exclusion criteria in place. The outcome measure was γH2AX expression in the tumors of the 3 mice used in this study.

### 2.3. Immunofluorescence

The GBM tissues obtained from the mice were fixed in 10% formalin and embedded in paraffin (formalin-fixed paraffin-embedded tissue). Four micron thick sections of the tissue were cut and subjected to immunofluorescence by following the recommended protocol (https://www.10xgenomics.com/support/cytassist-spatial-gene-expression/documentation/steps/tissue-staining/visium-cyt-assist-spatial-gene-expression-for-ffpe-deparaffinization-decrosslinking-immunofluorescence-staining-and-imaging, accessed on 1 March 2022). Briefly, the sections were deparaffinized and subjected to antigen retrieval followed by blocking and incubation with primary antibody (catalog number: 9718S, Cell Signaling Technology, Danvers, MA, USA) for 2 h followed by Texas red secondary antibody (catalog number: ab150080, Abcam, Cambridge, UK) for 1 h and counterstained using DAPI (catalog number: D9542, Sigma-Aldrich, Burlington, MA, USA). The entire section for each condition was then imaged on a Cytation (Gen5) using whole-slide scanning for the DAPI and Texas Red channels.

### 2.4. Spatial Transcriptomics

Following immunofluorescence, the tissue sections were subjected to spatial transcriptomics using the 10x Genomics platform at the Advanced Genomics Core (AGC), University of Michigan, employing the CytAssist technique as described in the Visium CytAssist Spatial Gene Expression for FFPE-Tissue Preparation Guide (CG000518, 10x Genomics, Pleasanton, CA, USA). Briefly, the tissue was permeabilized to release ligated probe pairs from the cells, which then bind to the spatially barcoded oligonucleotides present on the spots. The barcoded molecules are then used to generate a sequencing-ready library. The data was analyzed using 10x Genomics software resources (Space Ranger v 4.0 and Loupe Browser 8.1.0).

### 2.5. Annotation of the Spot γH2AX Positivity

Three researchers independently annotated all spots with at least 10 cells manually in all 3 sections, as well as using the MATLAB program. A total of 2785 spots in the no RT control tissue section, 1857 spots in the positive control (RT 30 min), and 2424 spots in the RT 6 h condition were annotated. Interobserver variability was compared using (a) average pairwise percentage agreement which is calculated as average of percentage agreement (% spots scored as positive by a pair of observers) + (% spots scored as negative by a pair of observers) between all pairs of observers and (b) Fliess’ Kappa statistic [11] (https://real-statistics.com/reliability/interrater-reliability/fleiss-kappa/, accessed on 1 March 2025) using ReCal, a inter-rater reliability calculator tool (https://dfreelon.org/utils/recalfront/recal3, accessed on 1 March 2025).

A Kappa value of 0 indicates no agreement, while a value of 1 indicates full agreement between observers.

### 2.6. Manual Annotation

γH2AX expression (red) had a punctate pattern within the nuclei (blue) (Figure 1A). A cell with a moderate to high intensity of Texas red was considered a positive cell. If 10% or more of the cells within the spot were positive, the spot was annotated as a positive spot. Spots containing fewer than 10 cells were eliminated.

### 2.7. MATLAB Program

Our program mainly utilizes the functions provided by MATLAB’s Image Processing Toolbox for its analysis. It was originally designed to detect γH2AX, a biomarker that labels DNA damage [12], on immunofluorescence, using the Texas Red fluorescence channel. However, the program can detect any fluorophore in the nucleus and can be adapted to include detection of cytoplasmic and membrane immunostaining.

Our program is designed to individually analyze the group of cells located under each spot on a tissue section. To focus analysis on the cells underneath each spot, the program applies a mask to the image. Since the size of the spots is fixed at 55 μM, no customization is necessary to perform masking. However, the spot’s coordinates are necessary as input into the program. These coordinates correspond to the same coordinates as those used for the barcodes in RNA extraction from the tissue. These coordinates or spot midpoints can be obtained from the Loupe browser, but they may also be user-defined (Appendix A provides the default settings for all user inputs).

Segmenting cells within spots is challenging due to the close proximity of nuclei in a 20×-magnification image, the standard magnification used in many spatial transcriptomic platforms. Many current programs are limited by their ability to define a cell’s nucleus when overlapping occurs [13,14,15]. This program is designed to define the boundaries of a nucleus using DAPI intensity. The program leverages the fact that the DAPI intensity is lower at the cell boundaries compared to their centers. The program first identifies all pixels above the user-defined threshold for DAPI intensity and deletes all pixels below the threshold (Figure 2). Following the first iteration, all blue regions completely surrounded by black and lying within the user-defined perimeter are considered as individual cells, and the information from them is recorded. These cells are then eliminated from the picture for the next iteration. For the second iteration, it eliminates all pixels below an increased threshold of DAPI (the amount of increase is referred to as the “range,” which is user-defined). This process continues until the upper limit of DAPI has been reached or until all the blue cells have been identified and the image is black. Each iteration gradually separates overlapping cells and stores the information for each cell. To ensure that most of the information is stored for overlapping cells, the user can manually define the perimeter of a cell. This perimeter refers to the average number of pixels that encompass a nucleus in an image. Providing an appropriate perimeter value prevents overlapping cells from being regarded as a single cell. Figure 2 illustrates the step-by-step processing of the image by the code.

To identify a certain biomarker within the nucleus, the user can set the minimum and maximum threshold values for an image’s red, green, and blue channels. This range indicates the intensity of fluorescence required for a pixel in an image to be considered positive for the biomarker and should be determined using proper biological controls. The program calculates the number of cells positive for the biomarker within a spot based on user input, which specifies the threshold of positive pixels required to consider a cell positive. The overall spot’s intensity value is based on the average pixel values within a spot using the fluorophore’s main color channel. For example, Texas Red mainly utilizes the image’s red channel. A mask is employed to use only the main channel’s positive pixels for the intensity calculation.

Based on the above information, the program analyzes each spot and produces the following output in the form of a table: the x and y coordinates of each spot’s center, the total number of cells and positive cells within each spot, the positivity value of each spot in percent form, each spot’s intensity value for the biomarker, and the final program’s call. The final program call will be either “positive” or “negative”, indicating whether the spot is positive or negative for the biomarker. This call is customizable by the user. It may be based on the spot’s intensity value, the percentage of positive cells present within the spot, or both values. The user has the choice to define the thresholds for each option.

The code has been deposited in GitHub (https://github.com/Radiation-oncology-wahl-lab/Spatial-transcriptomics-image-analysis/tree/main, accessed on 15 July 2024). Appendix A describes the default settings used in the MATLAB program.

## 3. Results

### 3.1. Different Experimental Conditions Yielded Tissues Showing Varying Expression of the Marker

The tumor analyzed in this study is glioblastoma, which is the most aggressive primary brain malignancy in adults. Radiation treatment is one of the mainstay treatment modalities for this cancer [16]. Glioblastoma is resistant to radiation treatment, as evidenced by tumor recurrence and a short median overall survival time of about 16 months [17,18]. Radiation treatment acts by inducing double-stranded DNA breaks [19]. The tumor’s ability to repair this damage quickly enables it to become resistant to treatment. For this study, we used glioblastoma cells grown in mouse brains, which were subjected to radiation treatment.

γH2AX is a phosphorylation marker that indicates the presence of unresolved double-stranded DNA (dsDNA) breaks (Figure 1A). Since radiation exposure induces dsDNA breaks, the expression of γH2AX is uniformly low in the unirradiated tissue (Figure 1B) and uniformly high in 30 min post RT tissue (Figure 1C). We demonstrate that in the 6 h post-RT tissue, some repair of the DNA damage has occurred in specific regions of the tumor, while damage persists in other regions (Figure 1D); hence, γH2AX expression is intermediate and non-uniform.

Spatial transcriptomic analysis platforms often provide gene expression data for spots, rather than individual cells. These spots vary in size depending on the analysis platform used, but are often on the order of 50–60 µm in diameter. In conditions where γH2AX staining is heterogeneous (i.e., 6 h after RT), some 55 µM spots were clearly negative (Figure 1E, blue circle) or positive (Figure 1E, red circle) for the γH2AX marker. Other spots (Figure 1E, yellow circle) exhibited a mixture of negative and positive cells, raising uncertainty about whether they should be labeled as positive or negative for γH2AX.

### 3.2. High Interobserver Variability Is Noted in Manual Annotation of Spots with Non-Uniform Marker Expression

To assess inter-observer variability in the analysis pipeline, three independent researchers annotated the spots based on agreed-upon criteria: moderate to high γH2AX expression within the nucleus indicated a positive cell, and 10% or more of positive cells within the spot were considered a positive spot. The tissues annotated were one unirradiated GBM tissue, which served as a negative control for γH2AX, one GBM tissue harvested 30 min after RT, which served as a positive control for γH2AX, and one GBM tissue harvested 6 h after RT, which had heterogeneous γH2AX expression. The annotations were compared, and interobserver variability was measured using percentage agreement and Fleiss’ Kappa statistic. A Kappa statistic of 0.8 or above is considered near-perfect agreement [11]. A percentage agreement of 80% or above is considered the minimum acceptable agreement [20].

In the unirradiated negative control tissue, 2785 spots were analyzed by three researchers. On average, each spot contained 42.1 cells (range: 11 to 88) as defined by the number of DAPI-stained nuclei. The tissue showed uniform low positivity of γH2AX expression. The percentages of spots annotated as positive by the three researchers are 9.4%, 9.4%, and 11.3%. There was high agreement among the three researchers on the annotation of individual spots. Indeed, the average agreement percentage is 98.42%, while the Kappa agreement statistic is 0.912, both of which indicate near-perfect agreement. Therefore, tissue showing uniform low positivity (Figure 1B) had good agreement among the three researchers.

In the 30 min post RT tissue, 1857 spots were analyzed by three researchers. On average, each spot contained 45.8 cells (range: 15 to 92) as defined by the number of DAPI-stained nuclei. The tissue showed uniform high positivity of γH2AX expression. The percentages of spots annotated as positive by the three researchers are 98.5%, 98.4%, and 98.4%. There was high agreement among the three researchers on the annotation of individual spots, with an average agreement percentage of 99.57% and a Kappa agreement statistic of 0.857, both of which indicate near-perfect agreement. Hence, the tissue showing uniform high γH2AX positivity showed good agreement among the three researchers.

In the 6 h post RT tissue, 2424 spots were analyzed by all three researchers. On average, each spot had 48.3 cells (range: 10 to 91) as defined by the number of DAPI-stained nuclei. The tissue showed non-uniform γH2AX expression. The percentages of spots annotated as positive by the three researchers are 43.2%, 65.4%, and 69.8%.

There was also low agreement among the three researchers on the annotation of individual spots, with an average agreement percentage of only 69.06% and a Kappa statistic of 0.345, both of which indicate poor agreement. Therefore, tissue showing non-uniform intermediate positivity for γH2AX was annotated differently by different researchers. Figure 1E depicts a spot (yellow circle) with high disagreement between researchers. Table 1 shows the extent of agreement (Fleiss’ Kappa value and average pairwise agreement scores) between observers in different experimental conditions with varying γH2AX immunopositivity.

As shown in Table 1, manual annotation of the spots resulted in significant differences in annotating the spots in the critical experimental condition, where subsequent RNA expression analysis heavily depends on the accuracy of spot annotation. The objective of spatial transcriptomics coupled with IF for γH2AX is to compare the RNA expression in γH2AX-positive spots with that in γH2AX-negative spots. The three observers generated heatmaps on Loupe Browser for differential RNA expression in γH2AX-positive vs. negative spots using their respective manual annotations of spots in the Rt 6 h tissue. The top 3 upregulated and downregulated genes were different when each observer used their respective annotations for analysis (Figure 3). Hence, it is essential to have an objective method for annotating the spots to ensure the reproducibility and reliability of the data analysis.

### 3.3. Development of a Quantitative Tool to Analyze Immunofluorescence in Multicellular Spots

To overcome interobserver variability in manual immunofluorescent annotation, we developed a quantitative analysis tool using MATLAB. Our program is designed to consider each spot (a cluster of cells) as a single image and perform nuclear segmentation in each spot individually (Figure 2).

Nuclear segmentation is performed through multiple iterations based on the intensity of DAPI and a user-defined range for iterations and the perimeter of the nucleus (Figure 4). Allowing the user to define these parameters enables customization of segmentation to suit different tissue types, which have varying cell densities and cell sizes.

Following segmentation, the Texas red intensity within the nuclei is measured, eliminating the chance of measuring RBC autofluorescence (red fluorescence without any underlying nucleus), as γH2AX. This reduces false-positive annotation of the spot. The program default is set to consider both Texas red intensity and the percentage of cells positive for γH2AX in the spot, annotating it as positive or negative. However, the user can choose to use only the intensity of the marker or only percentage positive cells to annotate the spots, depending on the immunostaining pattern of the marker in the specific tissue. Appendix A provides a detailed description of the user-defined parameters.

### 3.4. Automated Quantification of Immunofluorescence Intensity

Our MATLAB program output provided the number of DAPI-stained nuclei within the spot, the average intensity of Texas Red within the nuclei present in the spot, and the percentage of cells positive for Texas Red. We were able to objectively eliminate spots with too few cells (<10). The mean intensity of Texas red in the spots ranged from 55 to 202.92. The percentage of positive cells in the spots ranged from 0 to 100. Spots with average intensity above 60 and percentage positive cells >10% were annotated as positive spots.

In the no RT tissue showing uniform low expression of γH2AX, the MATLAB program annotated 10.1% spots as positive, which agreed well with manual annotation. In the 30 min post RT tissue with uniform high γH2AX expression, the MATLAB program annotated 98.5% spots as positive, which is similar to the total spots annotated as positive by the three researchers.

In the RT 6 h tissue with heterogeneous expression of γH2AX, the MATLAB program annotated 61.3% of the spots as positive, although this number could be altered by adjusting the intensity and the percentage of positive cells threshold. In spots with high disagreement, such as the central spot in Figure 1E which was annotated as positive by two researchers and negative by one researcher, the MATLAB program was able to objectively annotate it as negative because, although the average intensity of Texas red was 86.8, the percentage of positive cells was only 9.1% (Figure 5A(iii)).

Figure 5 illustrates different scenarios with high disagreement between observers, in which the MATLAB program was able to make an objective call. In Figure 5A(i), the spot was annotated as positive by two researchers due to high apparent Texas red intensity. However, the MATLAB program was able to identify that the red staining is not present over DAPI and is likely due to RBC autofluorescence, rather than true γH2AX positivity. It was then able to objectively classify it as a negative spot based on the criteria set by the researchers. The heatmap (Figure 5B) shows differentially regulated genes between γH2AX-positive and negative spots, as annotated by the MATLAB program. When the top 5 genes upregulated in γH2AX-negative spots are compared with those identified by different observers based on their manual annotation, the MATLAB call-based genes only matched with 2 of the genes identified by one observer. The top 5 upregulated genes identified by the other two observers did not match the top 5 genes identified using MATLAB annotation. This demonstrates that when attempting to identify candidate genes using IF-based marker annotation, it is crucial to have an objective tool to ensure reliable and reproducible results. Our MATLAB program enables such an objective assessment.

## 4. Discussion

Advances in spatial biology have yielded insights into tissue heterogeneity and cancer biology in a spatially relevant fashion. Several modalities for studying spatial transcriptomics are currently available [21,22,23], but they have the inherent limitation of not being able to investigate post-translational modifications and other epigenetic changes, such as histone modifications.

Post-translational modifications, such as the phosphorylation of histone H2A.x, are well-documented and have been shown to have a significant impact on the biology of several cancers. These modifications cannot be studied by measuring RNA expression, since the changes happen after the RNA has been translated into protein. Histone modifications such as trimethylation of H3K27 also cannot be detected using transcriptomics but can be identified using IF. Spatial protein expression can be studied using multiplex IF platforms with advanced image analysis programs like Akoya Biosciences [24,25]. However, this cannot be combined with transcriptomic analysis, and only a limited number of proteins can be studied at a time. This lack of combined transcriptomic and post-translational analysis can limit biological insights.

Combining immunofluorescence and spatial transcriptomics offers the opportunity to study both RNA and protein expression on the same tissue section. However, there are limited tools available to integrate spatial RNA sequencing data with IF images. This is especially relevant because current spatial transcriptomics platforms are predominantly based on capturing RNA from multicellular regions. The currently available image analysis tools like ImageJ and CellProfiler, and advanced bioinformatic tools like Seurat 5.3.0 based in R 4.5.1 and Scanpy 2.6.1 based in Python 3.12.6 which are used to analyze the spatial transcriptomics data, do not allow automated and objective image analysis of parameters like intensity of marker expression, percentage of marker positive cells within the multicellular spot and other image analysis parameters essential for downstream analysis. Such analyses need to be performed manually and are prone to interobserver variability.

Interobserver variability is widely reported in histopathological studies of cancer tissues, evaluating both routine histopathology images and images of immunostained sections [26,27,28]. Percentage agreement and Kappa statistics have been widely used to quantify inter-observer variability. A Kappa statistic of 0.8 and above is considered as near perfect agreement [11], which was seen in the evaluation of tissues with uniform low or uniform high expression of the marker, but not in the non-uniform marker expression condition. In the case of spatial transcriptomics, accurately and objectively annotating the spots based on the immunofluorescent image is essential for studying spatial RNA expression and drawing reliable and reproducible conclusions. Currently, there are no platforms that can quantitatively assess immunofluorescent images and annotate spots while complementing Cytassist. Our program is the first tool to accomplish this task.

Our program is designed to annotate multicellular spots, with varying marker expression within the spots, as either positive or negative based on user-defined criteria of “ground truth”. These criteria can be modified to suit the tissue type, experimental condition, and the marker expression pattern. Originally, the program was written to detect the presence of γH2AX; however, other nuclear markers of post-translational modifications, such as 5-methyl cytosine [29] and H3K27me3 (trimethyl H3K27), which are also prone to interobserver variability in manual annotation, could also be used [30]. The program enables researchers to adjust threshold values for all three color channels: red, green, and blue. The combination of these channels enables the detection of fluorescent markers of any color. For example, combining the red and green channels will allow the program to detect an orange fluorophore. The user can define this parameter, thereby extending the applicability of the tool to other IF images and multiplex IF images. The program’s nuclei stain detection is currently limited to DAPI or another blue stain as it is widely used as nuclear stain; however, this feature may be added to future versions, if required.

The output generated by this code is a table that can be exported as a .csv file, which can then be imported into existing browsers, such as Loupe Browser, used for spatial transcriptomics visualization and analysis. The table can also serve as input for more advanced single-cell RNA sequencing and spatial RNA sequencing analysis pipelines, such as the Seurat pipeline and Spacexr pipelines, which are based on R, as well as other pipelines based on Python, including Scanpy 2.6.1 and pySCENIC 0.12.1.

MATLAB code has several advantages over other tools, such as ImageJ and CellProfiler, because existing tools do not cater to the need of identifying multicellular spots. Furthermore, MATLAB offers image processing tools that can be customized to suit various tissue types. We leveraged this feature and customized our code to be flexible, enabling it to detect nuclei in tissues of varying nuclear densities, as well as nuclei with different shapes and sizes.

### Limitations

Our work also has some limitations. Our code currently only identifies nuclear immunostaining, as the segmentation is based on DAPI intensity. Since the essential step to tease apart overlapping cells mandates segmentation to be strictly limited to DAPI intensity, it cannot be easily modified to suit cytoplasmic or membrane staining in its current form. Another limitation is that this tool applies only to the annotation of circular multicellular spots. However, with modifications to defining parameters, the program can be modified to annotate the cells objectively based on IF.

## 5. Conclusions

In summary, our unique MATLAB program enables researchers to objectively assess immunofluorescent markers in conjunction with spatial transcriptomics, as performed on Cytassist, a commonly used spatial transcriptomics platform. It allows its user to accurately and efficiently analyze immunofluorescent nuclear biomarkers within spots without the need for laborious and subjective manual annotation.

## Figures and Tables

**Figure 1 cimb-47-00531-f001:**
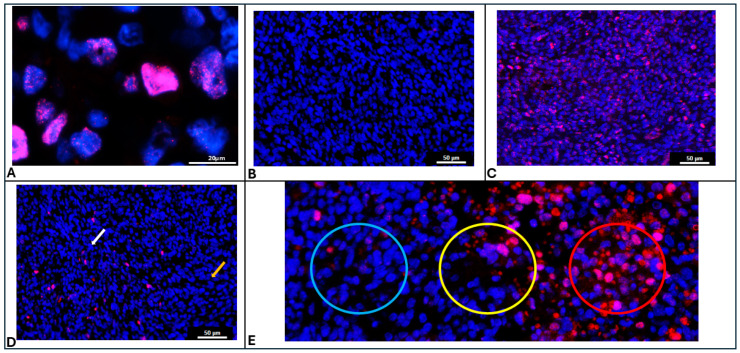
(**A**) Immunostaining pattern of γH2AX showing punctate patterns within the DAPI stained nucleus of irradiated cells (Blue: DAPI, Red: γH2AX). (**B**) Representative image of control mouse brain tumor without radiation exposure showing uniform absence of γH2AX immunostaining. (**C**) Representative image of mouse brain tumor obtained 30min after radiation exposure showing uniform high γH2AX immunostaining. (**D**) Representative image of mouse brain tumor obtained 6hr after radiation exposure showing non-uniform γH2AX immunostaining with regions of high expression (white arrow) and regions of low expression (yellow arrow). (**E**) Representative image showing Cytassist spots overlaid on γH2AX immunostained image depicting clearly negative spots (Blue circle), clearly positive spots (Red circle) and inconclusive spots (Yellow circle).

**Figure 2 cimb-47-00531-f002:**
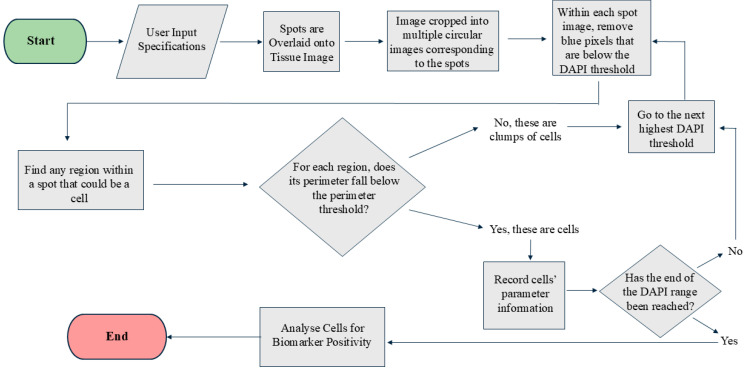
Step by step cell segmentation and analysis process of the MATLAB code.

**Figure 3 cimb-47-00531-f003:**
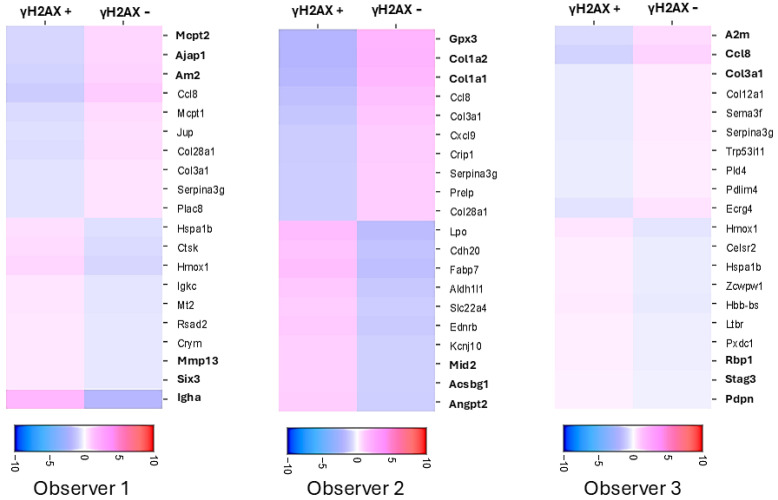
Heatmaps of differential gene expression between γH2AX positive and negative spots after annotation by different observers shows major differences in results. Scale bars show log2 fold change. The top 3 differentially expressed genes are different for different observers.

**Figure 4 cimb-47-00531-f004:**
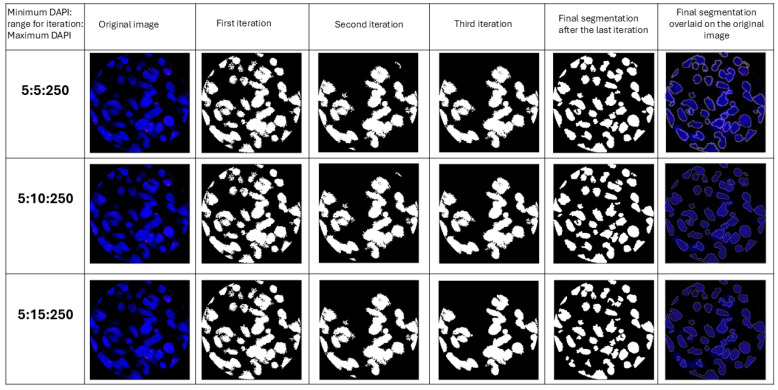
Images of sequential iterations and the final segmentation after the last iteration for different user defined ranges for iterations. Sequential iterations show how the code identifies cells and then excludes them from the image for the next iteration. The last column shows final masks overlaid on the original image.

**Figure 5 cimb-47-00531-f005:**
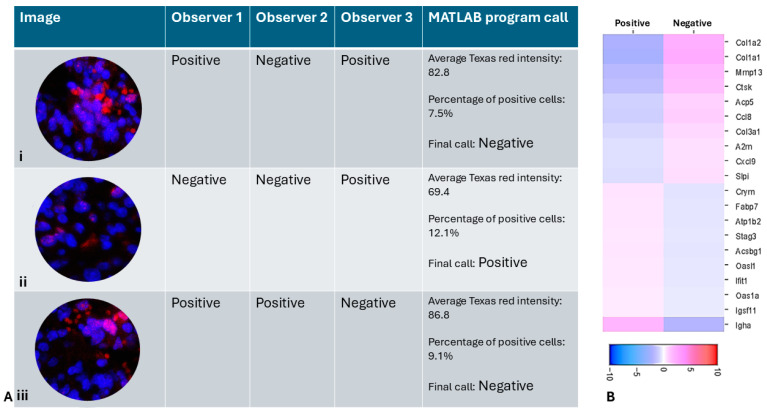
(**A**) (**i**–**iii**) Representative spots with high disagreement between observers and the MATLAB program call. The spots show nuclei stained with DAPI (blue) and γH2AX immunofluorescence (pink) and Red blood cell autofluorescence (red) (**B**) Heatmap of differential gene expression using MATLAB annotation showing significant difference compared to Figure 3. Scale bar shows log2 fold change.

**Table 1 cimb-47-00531-t001:** Agreement scores and Kappa statistic for 3 experimental conditions showing the least reliability in annotating the section with non-uniform γH2AX immunopositivity.

	Negative Control (No RT)	Positive Control (RT 30 min)	Critical Experimental Condition (RT 6 h)
No. of spots annotated	2785	1857	2424
Agreement between observers 1 and 2	98.06%	99.35%	73.93%
Agreement between observers 2 and 3	99.21	100%	57.47%
Agreement between observers 1 and 3	97.99%	99.35%	75.78%
Average pairwise agreement score	98.42%	99.57%	69.06%
Fliess’ Kappa	0.912	0.857	0.345

## Data Availability

The MATLAB code is deposited in GitHub (https://github.com/Radiation-oncology-wahl-lab/Spatial-transcriptomics-image-analysis/tree/main, accessed on 15 July 2024).

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
