# Peer review of "A New Tool to Decrease Interobserver Variability in Biomarker Annotation in Solid Tumor Tissue for Spatial Transcriptomic Analysis"

_cimb, 2025, doi:10.3390/cimb47070531_

Round 1
Reviewer 1 Report
Comments and Suggestions for Authors
This manuscript addresses a critical issue in spatial transcriptomics combined with immunofluorescence, interobserver variability in biomarker annotation, and proposes a MATLAB-based automated solution to standardize and improve the accuracy of spot annotations, specifically focusing on γH2AX as a DNA damage marker. The problem is highly relevant given the increasing adoption of spatial omics techniques in tumor biology research, and the authors' approach to developing an objective, automated tool is commendable.
Comments:
- While γH2AX is used as the example biomarker, it would be helpful to briefly mention other markers or applications where manual annotation variability is problematic, broadening the relevance.
- Clarify why MATLAB was chosen over other image analysis platforms (e.g., ImageJ, CellProfiler). Discuss advantages such as flexibility, customizability, or existing codebases.
- Emphasize that existing tools lack specificity or are not tailored for integrating immunofluorescence with spatial transcriptomics data, hence the necessity for developing a new solution
- Specify the image processing steps: method for nuclear segmentation (e.g., thresholding, watershed, machine learning), detection of immunofluorescence intensity thresholds, and handling of overlapping spots.
- Include parameter selection rationale: How were thresholds for positivity defined? Was there an adaptive or fixed threshold? How was this validated?
- Clarify if the program accounts for variation in cell size, nucleus shape, or staining intensity.
- Detail the dataset size (number of images/spots analyzed).
- Indicate if the MATLAB code is available publicly (e.g., GitHub).
- Mention computational requirements and system specifications, if relevant.
- Show example heatmaps or gene expression profiles for manual vs. automated annotation to illustrate impacts.
Reviewer 2 Report
Comments and Suggestions for Authors
Here the authors describe the development and validation of a MATLAB-based workflow designed to integrate immunofluorescence imaging with spot-based spatial transcriptomic data in solid tumor tissues. Focusing on γH2AX, a marker of DNA damage induced by radiotherapy in orthotopic murine glioblastoma, the authors first demonstrate that manual annotation of multicellular “spots” yields high interobserver variability when marker expression is heterogeneous, although nearly perfect agreement is observed in uniformly negative or positive tissues.
To overcome this subjectivity, they implemented a customizable, iterative nuclei-segmentation algorithm that leverages DAPI intensity thresholds to resolve overlapping cells, applies user-defined fluorescence intensity and positivity-percentage criteria to classify spots, and outputs spot coordinates, cell counts, average intensities, and binary calls. When benchmarked against manual calls, the automated tool reproduced annotations in control conditions and provided objective, reproducible classifications in critical heterogeneous samples, thereby standardizing downstream differential gene-expression analyses.
Major points:
The authors tackled a clear technological gap by quantitatively integrating IF and spatial transcriptomics—an intersection often hampered by mismatched spatial resolutions—and rigorously quantifies manual annotation errors using well-accepted reliability metrics.
The modular, parameter-driven design of the MATLAB programming includes different nuclear biomarkers, allows users to tailor threshold settings to tissue-specific staining patterns, and is openly accessible via GitHub, which together promote transparency and reproducibility.
The iterative segmentation strategy addresses the challenge of resolving densely packed nuclei, and the authors show that their automated approach preserves meaningful biological contrasts in downstream analyses.
Minor points:
Some limitations should be adressed in the discussion part of the manuscript. The tool’s applicability is currently confined to nuclear markers and circular spot geometries typical of one commercial platform, limiting its direct use for cytoplasmic or membrane stains and for technologies featuring non-circular capture regions.
Furthermore, the validation is restricted to a single physiological context (γH2AX labeling in irradiated glioblastoma) so its performance across diverse tissues, markers and imaging conditions remains to be demonstrated.
Moreover, other aspects should be discussed in details, as the authors do not report on computational efficiency or integration with established bioinformatics pipelines, leaving questions about scalability in large-scale studies.
